



# A microbially-driven and depth-explicit soil organic carbon model constrained by carbon isotopes to reduce equifinality

Marijn Van de Broek[1], Gerard Govers[2], Marion Schrumpf[3], and Johan Six[1]

[1]Department of Environmental Systems Science, ETH Zurich, Zürich, Switzerland
[2]Division of Geography and Tourism, Department of Earth and Environmental Sciences, KU Leuven, Leuven, Belgium
[3]Department for Biogeochemical Processes, Max Planck Institute for Biogeochemistry, Jena, Germany

**Correspondence:** Marijn Van de Broek (Marijn.vandebroek@usys.ethz.ch)

**Abstract.**

Over the past years, microbially-driven models have been developed to improve simulations of soil organic carbon (SOC), and have been put forward as an improvement to assess of the fate of SOC stocks under environmental change. While these models do include a better mechanistic representation of SOC cycling in comparison to cascading reservoir-based approaches, the complexity of these models implies that data on SOC stocks are insufficient to constrain the additional model parameters. In this study, we constructed a novel depth-explicit SOC model (SOILcarb) that incorporates multiple processes influencing the $\delta^{13}$C and $\Delta^{14}$C values of SOC and assessed if including data on the $\delta^{13}$C and $\Delta^{14}$C value of SOC during parameter reduces model equifinality, the phenomenon that multiple parameter combinations lead to a similar model output. To do so, we used SOILcarb to simulate depth profiles of total SOC and its $\delta^{13}$C and $\Delta^{14}$C values. The results show that when the model is calibrated based on only SOC stock data , the residence time of subsoil organic carbon (OC) is not simulated correctly, thus effectively making the model of limited use to predict SOC stocks driven by, for example, environmental changes. Including data on $\delta^{13}$C in the calibration process reduced model equifinality only marginally. In contrast, including data on $\Delta^{14}$C in the calibration process resulted in simulations of the residence time of subsoil OC consistent with measurements, while reducing equifinality only for model parameters related to the residence time of OC associated with soil minerals. Multiple model parameters could not be constrained even when data on both $\delta^{13}$C and $\Delta^{14}$C were included. Our results show that equifinality is an important phenomenon to consider when developing novel SOC models, or when applying established ones. Reducing uncertainty caused by this mechanism is necessary to increase confidence in predictions of the soil – climate feedback in a world subject to environmental change.

## 1 Introduction

Soils are an important component of the global carbon cycle, storing a vast amount of organic carbon (OC) (Scharlemann et al., 2014; Ciais et al., 2013). However, it is often overlooked that about 77 % of the soil organic carbon (SOC) in the upper 3 m of soil is present below a depth of 0.3 m (Lal, 2018). Furthermore, while topsoil (< 0.3 m depth) OC is generally characterized by average residence times ranging from years to decades (Baisden et al., 2013; Schrumpf and Kaiser, 2015), subsoil (> 0.3 m depth) OC typically has residence times up to centuries or millennia (Balesdent et al., 2018; Luo et al., 2019). Despite these



long residence times, subsoil OC is likely to play an important role in the climate – soil carbon feedback, as subsoil OC has been shown to be susceptible to losses upon soil warming (Hicks Pries et al., 2017; Soong et al., 2021; Jia et al., 2019). Thus, a correct representation of subsoil OC in biogeochemical models is necessary to make accurate predictions about climate – soil carbon feedbacks and land use changes for the coming decades (He et al., 2016; Wang et al., 2019).

Classical models simulating subsoil OC dynamics are based on conceptual pools with intrinsic turnover times, with the sim-
ulated carbon generally cascading along a sequence of model pools. The rate of OC turnover is generally calculated as a function of abiotic factors and the chemistry of organic compounds. Such models have been developed as stand-alone models (e.g. Elzein and Balesdent, 1995; Ota et al., 2013; Wang et al., 2015), or have been incorporated in ecosystem models (Camino-Serrano et al., 2018; Koven et al., 2013). These biogeochemical models have been criticized, as they do not incorporate the emerging understanding of the controls on soil organic carbon (SOC) dynamics (Blankinship et al., 2018; Schmidt et al., 2011;
Dungait et al., 2012; Bradford et al., 2016). For example, these models do not explicitly simulate soil microbes, which are both decomposers and important precursors of stabilized SOC (Denef et al., 2010; Kästner and Miltner, 2018; Kögel-Knabner, 2002; Six et al., 2006), and organo-mineral associations, which protect SOC from decomposition (Kleber et al., 2015, 2021; Sollins et al., 1996). In addition, model pools with a strong inherent recalcitrance, such as the 'passive pool' in models such as Daycent (Parton et al., 1987) and the 'humified organic matter pool' in RothC (Coleman et al., 1997), are assumed to be
the result of humification, a theory that is being considered flawed and obsolete (Kleber and Lehmann, 2019; Lehmann and Kleber, 2015).

As a reaction to this emerging understanding of the controls on SOC dynamics over the past decades, several mechanistic, microbially-driven models simulating depth profiles of SOC have been developed (e.g., Ahrens et al., 2015, 2020; Dwivedi et al., 2017; Riley et al., 2014; Yu et al., 2020; Zhang et al., 2021). These models share multiple characteristics, such as the
explicit representation of soil microbes and the protection of SOC by association with soil minerals. Moreover, the increasing residence time of SOC with soil depth is simulated as an emerging function of biotic and abiotic soil properties (Ahrens et al., 2020). This improves the mechanistic representation of SOC dynamics in these models compared to first-order decay models, which generally use an exponentially decreasing parameter with depth to force the decreasing processing rate of soil organic carbon along the soil profile (Koven et al., 2013; Wang et al., 2015, 2020).

This new generation of SOC models is characterised by an increase in model complexity and in the number of model param-eters (Campbell and Paustian, 2015; Lawrence et al., 2009). This is an important consideration, as an increase in parameter uncertainty can outweigh model improvements due to a better mechanistic description of the system, thereby increasing the overall model error (Van Rompaey and Govers, 2002). Finding an optimal balance between errors related to process represen-tation and data availability is of major importance to create confidence in model outputs (Schindler and Hilborn, 2015). A too
complex model with respect to the availability of data can result in multiple combinations of parameter values that lead to a near-optimal solution, so-called 'behavioural models'. This phenomenon has been referred to in literature by multiple terms, such as equifinality (e.g. Beven, 2006, 1993), non-uniqueness (Beven, 2006) or non-identifiability of model parameters (Brun et al., 2001; Sierra et al., 2015). Equifinality is a major issue for models simulating hydrology (e.g. Beven and Freer, 2001), soil organic carbon (e.g. Sierra et al., 2015; Braakhekke et al., 2013; Marschmann et al., 2019), soil nitrogen (e.g. Schulz et al.,



1999) and ecosystem models in general (e.g. Luo et al., 2009; Tang and Zhuang, 2008). In the present article, we use the
term *parameter equifinality* for this phenomenon. In addition, the term *overparameterisation* is used for the situation when the
number of model parameters is too large with respect to the available observational support for the processes represented by
these parameters.

One way to reduce parameter equifinality is to include additional constraints on model parameter values during the calibration
process (Braakhekke et al., 2014). This has been done for multiple classical SOC models, by simulating, in addition to total
SOC, depth profiles of the ratio of stable carbon isotopes ($\delta^{13}$C) (Amundson and Baisden, 2000; van Dam et al., 1997; Poage
and Feng, 2004), radioisotopes ($\Delta^{14}$C) (Baisden and Parfitt, 2007; Jenkinson and Coleman, 2008; Koven et al., 2013; Tifafi
et al., 2018; Braakhekke et al., 2014) or a combination of both (Wang et al., 2020; Baisden et al., 2002), or $^{210}$Pb (Braakhekke
et al., 2013). Some mechanistic models also simulate the behaviour of $^{14}$C (Ahrens et al., 2015, 2020; Dwivedi et al., 2017;
Yu et al., 2020). While it has been shown that simulating these additional variables puts meaningful constraints on parameter
values, measurements of the necessary data is costly and hence such data are not widely available. On the other hand, $\delta^{13}$C iso-
tope ratios can be rapidly and relatively cheaply measured. While these isotopes do not decay radioactively like $^{14}$C, multiple
processes that take place over decadal to centennial timescales influence depth patterns of the $\delta^{13}$C value of SOC. For example,
it has been shown that long-term changes in the $\delta^{13}$C value of vegetation (Keeling, 1979; Schubert and Jahren, 2015) influence
the $\delta^{13}$C value of SOC along the depth profile considerably (Paul et al., 2019). In addition, there is evidence that microbial
necromass, which constitutes up to 50 % of SOC (Rumpel and Kögel-Knabner, 2010; Wang et al., 2021; Angst et al., 2021), is
generally enriched in $^{13}$C compared to their substrate (Dijkstra et al., 2006; Gleixner et al., 1993; Miltner et al., 2004). Explic-
itly simulating the fate of microbial necromass, and its $^{13}$C isotopes, therefore has the potential to better constrain the rate of
the formation of stabilized microbial necromass in soils (Šantrůčková et al., 2018). Lastly, as above- and belowground C inputs
to the soil generally have different $\delta^{13}$C values (Bowling et al., 2008; Werth and Kuzyakov, 2010), simulating the $\delta^{13}$C value of
these inputs has the potential to better constrain the vertical mixing of OC from different sources along the soil profile. To the
best of our knowledge, the potential of using $^{13}$C isotope data to further constrain the parameter values of a microbially-driven
and depth-explicit SOC model has to date not been explored.

Therefore, the aim of this study is to assess to what extent the inclusion of simulating the $\delta^{13}$C and $\Delta^{14}$C values of SOC, in ad-
dition to SOC itself, allows to better constrain model parameter values of a microbially-driven and depth-explicit SOC model;
thereby reducing model equifinality. To do so, we constructed a novel depth-explicit SOC model that incorporates multiple
processes that influence the $\delta^{13}$C and $\Delta^{14}$C value of SOC. We hypothesized that (1) calibrating a microbially-explicit model
using only carbon stock data results in substantial parameter equifinality and (2) underestimates the residence time of subsoil
OC (following He et al. (2016)), while (3) using simulated depth profiles of $\delta^{13}$C or $\Delta^{14}$C, or a combination of both, as addi-
tional constraints on parameter values will narrow the range in optimised parameter values which result in behavioural models
(i.e. a model solution that cannot easily be rejected). As the $\delta^{13}$C value of SOC is generally not simulated in mechanistic SOC
models, we also discuss the effect of different mechanisms on simulated depth profile of $\delta^{13}$C.



## 2 Materials and Methods

### 2.1 The SOILcarb model

This section provides a brief description of the main processes simulated in SOILcarb (**S**imulation of **O**rganic carbon and its **I**sotopes by **L**inking **ca**rbon dynamics in the **r**hizosphere and **b**ulk soil). A detailed model description and overview of the equations is provided in the Supplementary Information, as well as an overview of the state variables (Table S5) and model parameters (Table S6). The presented version of SOILcarb has been developed to simulate depth profiles of OC dynamics in natural forest soils. The vertical soil profile is simulated down to 1 m depth, for layers with an increasing thickness with depth.

SOILcarb has been programmed in R (R Core Team, 2024), with the differential equations regulating the flows of OC being solved using the *lsodes* solver from the DeSolve package (Soetaert et al., 2010). For the presented simulations, the model was run for a period of 15,000 years up to the year 2004. It is noted that the current model does not include the effect of temperature and soil moisture, limiting the model to predict SOC and its isotopic signature under steady-state environmental conditions.

### 2.1.1 Simulation of organic carbon dynamics

SOILcarb is divided into three compartments: (1) litter layer, (2) rhizosphere and (3) bulk soil (Fig. 1). The litter layer is simulated spatially separated from the soil compartments. The rhizosphere and bulk soil compartments are used to conceptually separate the parts of the soil where most OC inputs occur and OC cycles relatively fast (the rhizosphere) from the zone where available OC for microbes is relatively limited due to mineral protection and where OC cycles relatively slow (the bulk soil). Inputs of OC in the litter layer originate from litterfall, which is separated into particulate OC ($C_{POC-l}$) and dissolvable OC

($C_{DOC-l}$). Depolymerisation and microbial uptake of $C_{POC-l}$ and $C_{DOC-l}$ are simulated using the equilibrium chemistry approximation (Tang and Riley, 2013) (the model assumes that DOC needs to be depolymerised before uptake, as a considerable portion of DOC is generally not bio-available (Risse-Buhl et al., 2013; Shen et al., 2015; Andreasson et al., 2009)), while microbial turnover is simulated as a logistic growth process (Georgiou et al., 2017). Microbial OC uptake in all compartments is reduced based on a fixed carbon use efficiency (CUE), with the remaining OC being transformed to $CO_2$. Carbon from the

litter layer is transferred to the bulk soil through bioturbation of $C_{POC-l}$ and leaching of $C_{DOC-l}$.

In the rhizosphere, OC inputs are separated into (1) rhizodeposits, providing bio-available OC ($C_{bioav-r}$) to the soil, and (2) root turnover, providing particulate OC ($C_{POC-r}$) to the soil. Depolymerisation of $C_{POC-r}$ to $C_{bioav-r}$ is simulated using reverse Michaelis-Menten kinetics, whereby the rate of depolymerisation is modified based on the ratio of microbes in the rhizosphere ($C_{mic-r}$) to $C_{POC-r}$ (see Supplementary Information). Uptake of $C_{bioav-r}$ by $C_{mic-r}$ is simulated using forward Michaelis-Menten

kinetics, whereby the rate of OC uptake is modified based on the ratio of $C_{mic-r}$ to $C_{bioav-r}$. Following microbial turnover in the rhizosphere (simulated using a logistic growth function), the soluble portion of microbial cells (the cytoplasm) is transferred back to $C_{bioav-r}$, while the non-soluble portion of microbial cells is transferred to the DOC pool in the bulk soil ($C_{DOC-b}$). A fixed portion of $C_{bioav-r}$ is transferred to $C_{DOC-b}$, to allow the direct adsorption of root-derived OC on soil minerals, without first passing through a soil microbe. From $C_{DOC-b}$, OC can be either protected by adsorption on soil minerals ($C_{min-b}$), rendering it

inaccessible to microbial uptake, or be taken up by microbes in the bulk soil ($C_{mic-b}$). Competition for $C_{DOC-b}$ between minerals



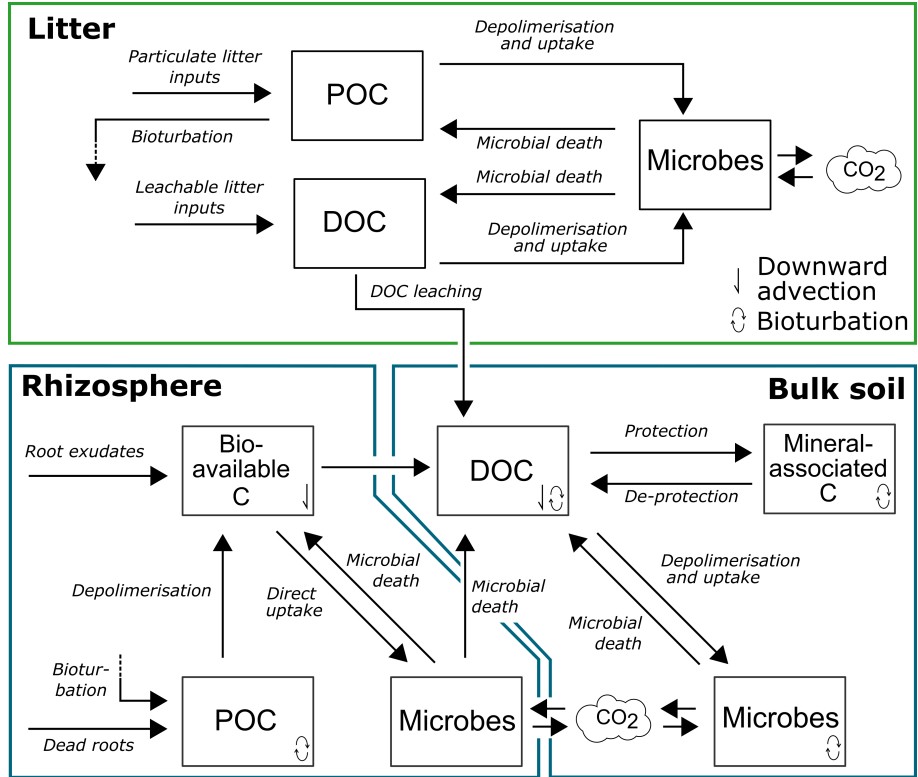

**Figure 1.** Conceptual model of SOILcarb showing the model pools and fluxes of organic carbon in the litter layer, rhizosphere and bulk soil. *POC* = particulate organic matter; *DOC* = dissolvable organic matter.

and microbes is simulated using the equilibrium chemistry approximation (Tang and Riley, 2013). De-protection of $C_{min-b}$ is simulated as a first-order process. Vertical transport of OC along the soil profile occurs as (1) bioturbation, simulated as a diffusion process (for $C_{POC-r}$, $C_{DOC-b}$, $C_{min-b}$ and $C_{mic-b}$), and (2) leaching, simulated as an advection process (for $C_{bioav-r}$ and $C_{DOC-b}$). It has been shown that the rate of de-protection of mineral-associated OC is influenced by root exudates (Keiluweit et al., 2015). Therefore, the simulated rate of de-protection of OC from minerals is a function of the portion of the soil occupied by the rhizosphere, calculated following Finzi et al. (2015). In addition, the amount of mineral surfaces available for the protection of OC is scaled according to the rhizosphere volume (i.e., the larger the rhizosphere volume, the larger the amount of minerals which are in contact with OC originating from the rhizosphere).

### 2.1.2 Simulation of $\delta^{13}$C and $\Delta^{14}$C of organic carbon

In SOILcab, fluxes of $^{13}$C and $^{14}$C between model pools follow fluxes of $^{12}$C. The model first calculates fluxes of $^{12}$C between pools and subsequently uses the ratio of $^{12}$C leaving every pool to the total amount of $^{12}$C of the respective pools to calculate



how much $^{13}$C and $^{14}$C leave every pool, based on the respective $^{13}$C/$^{12}$C and $^{14}$C/$^{12}$C values of the pools. The model parameters are thus defined based on the $^{12}$C content of every pool.

The simulated processes that affect temporal variations in the $\delta^{13}$C value of SOC are (1) annual changes in the $\delta^{13}$C value
of atmospheric $CO_2$, directly affecting the $\delta^{13}$C value of vegetation, (2) the effect of atmospheric $CO_2$ concentration on kinetic fractionation against $^{13}$C during plant photosynthesis, (3) differences in the $\delta^{13}$C value of aboveground plant biomass, belowground biomass and rhizodeposits, and (4) heterotrophic $CO_2$ assimilation by soil microbes. The same processes affect the temporal variation of the $\Delta^{14}$C of SOC, in addition to radioactive decay. Note that no preferential microbial mineralisation of $^{12}$C relative to $^{13}$C is simulated, as consistent empirical evidence for this is lacking (Boström et al., 2007; Ehleringer et al., 2000).

The difference in the $\delta^{13}$C value between atmospheric $CO_2$ and aboveground biomass is calculated for every time step as the sum of a fixed and variable component:

$$diff^{13}C_{atm-leaf}(t) = diff_{fixed} + diff_{variable}(t) \qquad (1)$$

Where $diff_{fixed}$ is a constant representing a fixed and user-provided difference in $\delta^{13}$C between atmospheric $CO_2$ and aboveground biomass, and $diff_{variable}(t)$ represents the effect of atmospheric $CO_2$ concentration on kinetic fractionation against $^{13}$C during photosynthesis for every time step (see supplementary information Sect. 1.6.3).

The value of $diff_{fixed}$ is provided by the user for the last simulation year, and is assumed to be constant throughout the simulation. For every simulated time step, the $\delta^{13}$C value of aboveground biomass is subsequently calculated from a long-term series of average annual $\delta^{13}$C values of atmospheric $CO_2$, which was compiled from Schmitt et al. (2012), Bauska et al. (2015) and Graven et al. (2017) (see Fig. S2). The annual $\Delta^{14}$C value of OC inputs from aboveground biomass is determined from a compiled series of annual average $\Delta^{14}$C values of atmospheric $CO_2$ (from Reimer et al. (2013), Hua et al. (2013) and Hammer and Levin (2017), see Fig. S3), combined with the assumption that during photosynthesis, the fractionation against $^{14}CO_2$ is twice that of against $^{13}CO_2$.

The second simulated mechanism affecting temporal changes in the $\delta^{13}$C and $\Delta^{14}$C values of OC inputs to the soil is the effect of the atmospheric $CO_2$ concentration on kinetic fractionation against $^{13}$C during photosynthesis. This is based on observations showing that for C3 plants, the magnitude of fractionation against $^{13}$C during photosynthesis increases with increasing atmospheric $CO_2$ concentration (Keeling et al., 2017; Schubert and Jahren, 2012, 2015). It has been shown that accounting for this mechanism improves simulations of depth profiles of the $\delta^{13}$C value of SOC (Paul et al., 2019). The model simulates a linear effect of the atmospheric $CO_2$ concentration on kinetic fractionation against $\delta^{13}$C during photosynthesis (Keeling et al., 2017):

$$diff_{variable}(t) = ([CO_2](t_{end}) - [CO_2](t)) \cdot S \qquad (2)$$

Where $[CO_2](t_{end})$ is the atmospheric $CO_2$ concentration (ppm) in the last simulated calendar year (*tend*), $[CO_2](t)$ is atmospheric $CO_2$ concentration in every other simulated year *t* and *S* represents the change in fractionation against $^{13}$C by plants





per unit change in atmospheric $CO_2$ concentration (‰ ppm$^{-1}$; Schubert and Jahren (2015)). The value of $S$ was fixed at 0.014

‰ ppm$^{-1}$, following Keeling et al. (2017).

With respect to variations in the $\delta^{13}C$ value of SOC with depth, a first simulated process is caused by differences in $\delta^{13}C$ between aboveground biomass, roots, and rhizodeposits. The $\delta^{13}C$ value of aboveground biomass is calculated for every time step using Eq. 1, while the $\delta^{13}C$ values of roots and rhizodeposits are calculated using user-defined differences in $\delta^{13}C$ between aboveground vegetation on the one hand, and roots and rhizodeposits respectively, on the other hand (see supplementary infor-

mation Sect. 1.6.4). The second mechanism is heterotrophic $CO_2$ assimilation by soil microbes (Šantrůčková et al., 2005, 2018; Nel and Cramer, 2019; Akinyede et al., 2020, 2022). In our model simulations, we assumed that soil microbes derive 1.1 % of their OC from heterotrophic $CO_2$ assimilation, as quantified for a soil in Hainich National Park by Akinyede et al. (2020) (see supplementary information Sect 1.3.1). To simulate the effect of the $\delta^{13}C$ and $\Delta^{14}C$ values of soil $CO_2$ along the depth profile on the microbial $\delta^{13}C$ and $\Delta^{14}C$ values due to heterotrophic $CO_2$ assimilation, depth profiles of the $\delta^{13}CO_2$ and $\Delta^{14}CO_2$ were

simulated using a one-dimensional $CO_2$ diffusion model (Amundson and Davidson, 1990; Cerling, 1984; Goffin et al., 2014) (see supplementary information Sect. 1.4).

## 2.2 Study site

The model was applied to a deciduous forest site in Hainich National Park (Germany; 51°04'N, 10° 27' E), using data from Schrumpf et al. (2013). The soil is an eutric Cambisol with a sand, silt and clay content of ca. 3, 38 and 59 %, respectively

(Schrumpf et al., 2011, 2013). Soil samples were collected in 2004 in three replicates for depth intervals of 0-5, 5-10, 10-20, 20-30, 30-40, 40-50 and 50-60 cm. Using density fractionation on 2 mm sieved soil, the amount of OC in (1) the free light fraction (referred to as particulate organic carbon (POC)), (2) the occluded light fraction and (3) the heavy fraction (referred to as mineral-associated organic carbon (MAOC)) was obtained. As SOILcarb does not simulate aggregate dynamics, the total amount of measured OC was reduced by the amount of OC in the occluded light fraction. Measured values of total OC, POC,

MAOC, and $\delta^{13}C$ and $\Delta^{14}C$ of the POC and the MAOC fractions were used for model calibration purposes. More information about the study site and data processing is provided in Schrumpf et al. (2011, 2013).

The annual amount of litterfall and root production at the study site was obtained from Kutsch et al. (2010), who, using measurements between 2000 and 2007, obtained an annual average rate of aboveground litter input of 209 ± 14 g C m$^{-2}$ yr$^{-1}$ and root production of 232 ± 15 g C m$^{-2}$ yr$^{-1}$. The annual production of rhizodeposit OC was calculated by multiplying the

annual root carbon production by 0.4, the median ratio of net rhizodeposition to the root biomass from a meta-analysis from forest soils by Pausch and Kuzyakov (2018). We note that this number is likely to be an underestimation because it does not account for post-rhizodeposition losses. This led to a total annual belowground OC flux of 324 g C m$^{-2}$, of which 92 g C m$^{-2}$ as rhizodeposits (29 %). The root and rhizodeposit inputs were distributed over the upper 1 m of the soil following the asymptotic nonlinear model of Gale and Grigal (1987) (see supplementary information Sect. 1.3.3). To calibrate the amount of OC in the

simulated litter layer, measurements of the litter and organic layer by Schrumpf et al. (2013) were combined (580 g C m$^{-2}$). The depth profile of OC stocks in the POC and MAOC fractions were obtained by combining measured OC concentrations for the respective fractions with the bulk density of the same depth layers (Schrumpf et al., 2013).





The $\delta^{13}$C value for the litter layer was calculated to be 0.1 ‰ lower than the average $\delta^{13}$C value of leaves (see below), following Knohl et al. (2005), resulting in a $\delta^{13}$C value of -29.2 ‰. The $\Delta^{14}$C value of the litter layer (98.7 ‰) was measured in 2004 by Schrumpf et al. (2013). The $\delta^{13}$C value of root inputs in 2004 was derived from the average measured $\delta^{13}$C value of POC below 0.2 m depth (-27.8 ‰), assuming that this POC is mostly derived from roots. The $\delta^{13}$C value of aboveground vegetation was derived from the measured difference of 1.5 ‰ in the $\delta^{13}$C value between roots and leaf area index-weighted leaves at the same site (Knohl et al., 2005), resulting in a $\delta^{13}$C value for aboveground biomass in 2004 of -29.3 ‰. As measurements of the $\delta^{13}$C value of root exudates were not available, a range of values was tested and the resulting $\delta^{13}$C value of SOC was compared to measured values, resulting in an optimal $\delta^{13}$C value for root exudates of -28.9 ‰. The heavier isotopic signature of root exudates compared to leaves is in line with the fact that root exudates are composed of sugars, amino acids and organic acids, among other chemical compounds (Pinton et al., 2007), which are enriched in $^{13}$C compared to bulk leaves (Bowling et al., 2008).

## 2.3 Parameter optimisation

### 2.3.1 Litter parameter optimisation

Parameter optimization was performed using the differential evolution (DE) algorithm from the *DEoptim* package in R (Mullen et al., 2011; Ardia et al., 2011), an evolutionary optimization algorithm to find optimal global parameter values in a complex multidimensional parameter space. Parameter optimization was performed separately for the litter and soil layers. For the litter layer, 3 parameter values were optimized: the half-saturation constants for POC depolymerisation ($K_{m\_POC-l}$) and DOC depolymerisation and uptake ($K_{m\_DOC-l}$), in addition to the maximum rate for both of these processes ($V_{max\_l}$). As no information on the distribution of the total amount of litter C between the simulated model pools ($C_{POC\text{-}l}$, $C_{DOC\text{-}l}$ and $C_{mic\text{-}l}$) was present, and the focus of the present study is on SOC dynamics, the amount of measured OC in the litter layer was assumed to be distributed as follows: 33 % as DOC, 66 % as POC and 1 % as microbial C. The error for simulations of the litter layer was calculated by summing the squared relative errors for the individual litter pools and isotopic constraints:

$$\epsilon_{lit} = \sum_{i=1}^{5} \left( \frac{meas_i - mod_i}{meas_i} \right)^2 \tag{3}$$

Where $\epsilon_{lit}$ is the total error for the litter layer (unitless), *meas* are the measured pools, *mod* are the modelled pools and *i* refers to the calibrated model pool ($C_{POC\text{-}l}$, $C_{DOC\text{-}l}$ and $C_{mic\text{-}l}$, $\delta^{13}$C$_{lit}$ and $\Delta^{14}$C$_{lit}$, where the latter two refer to the $\delta^{13}$C and $\Delta^{14}$C values of total litter OC).

### 2.3.2 Soil parameter optimisation

During parameter optimization, the measured POC fraction was compared to the modelled C$_{POC\text{-}r}$ pool, while the measured MAOC pool was compared to the simulated OC in the bulk soil, referred to here as $C_{bulk}$ (i.e., the sum of $C_{min-b}$, $C_{DOC-b}$ and $C_{mic-b}$). To assess the effect of isotopic constraints ($\delta^{13}$C and $\Delta^{14}$C) on optimized parameter values of SOILcarb, the model parameters were optimised with 4 different scenarios:



– Scenario 1: optimisation with OC data only. The optimised model pools are the amount of OC in POC ($C_{POC-r}$) and in the bulk soil ($C_{bulk}$).

– Scenario 2: optimisation with OC and $\delta^{13}$C data. The optimised model pools are $C_{POC-r}$, $C_{bulk}$, $\delta^{13}C_{\text{POC-r}}$ and $\delta^{13}C_{\text{bulk}}$.

– Scenario 3: optimisation with OC and $\Delta^{14}$C data. The optimised model pools are $C_{POC-r}$, $C_{bulk}$, $\Delta^{14}C_{\text{POC-r}}$ and $\Delta^{14}C_{\text{bulk}}$.

– Scenario 4: optimization with OC, $\delta^{13}$C and $\Delta^{14}$C data. The optimised model pools are $C_{POC-r}$, $C_{bulk}$, $\delta^{13}C_{\text{POC-r}}$, $\delta^{13}C_{\text{bulk}}$, $\Delta^{14}C_{\text{POC-r}}$ and $\Delta^{14}C_{\text{bulk}}$.

In addition, parameter sets were rejected during the calibration if the simulated model outcome did not meet the following criteria: (1) the amount of $C_{bioav-r}$ has to be smaller than the amount of $C_{POC-r}$ and (2) the total mass of OC in soil microbes (i.e., the sum of $C_{mic-r}$ and $C_{mic-b}$) cannot exceed 5 % of total simulated SOC. The errors of the respective pools were calculated as squared relative errors, similar to Eq. 3. The errors for the same model pool along the depth profile were summed to obtain the total error for every pool.

In a first step, we selected 11 parameters which were deemed to be most critical and for which no measured values nor reasonable estimates were available in the literature (Table S1). After optimization of these parameters, a sensitivity analysis was performed (see Sect. 2.5.1). This led to the identification of 9 model parameters that were optimised (Table S2) under the 4 scenarios outlined above.

Two model parameters were not retained for calibration. The first one is the rate of DOC advection ($\nu$), because of its limited sensitivity (Fig. S5). The second parameter is the e-folding depth of bioturbation ($z_b$), to avoid correlations with the biodiffusion coefficient ($D_b(0)$) despite having an influence on model results (Fig. S5). We note that although the parameters $V_{max,POC-r}$ and $V_{maxU,mic-r}$ had a minimal influence on model outcomes, these were retained to assess if parameters in the rhizosphere were prone to equifinality. For the optimisation of parameter values using the DE algorithm, 300 iterations were run. During each iteration 180 parameter combinations were tested, resulting in a total of 54,000 model runs per optimisation scenario.

## 2.4  Assessment of parameter equifinality

After parameter optimisation using the DE algorithm, equifinality of the optimized parameters was analyzed. Multiple methods are available to this end, such as GLUE (Beven and Binley, 1992) and Bayesian approaches (e.g. Vrugt, 2016), but these methods require prior information on the parameter value distribution to perform optimally. As this information was not available for the optimised parameters, an alternative approach was developed.

The DE algorithm efficiently explores the multi-dimensional parameter space by proposing new sets of parameter combinations during every iteration, based on previously generated parameter sets. To assess if multiple parameter combinations resulted in behavioural models, we kept track of all tested sets of parameter values during the optimisation procedure. For every calibration scenario, this resulted in 54,000 non-unique parameter sets that were generated in the parameter space. In



a next step, the model was run using the unique parameter combinations, and the results and respective model errors were stored. The parameter sets resulting in the 10 % lowest errors were retained, and were considered to be behavioural models, after visually assessing that the model results were within the uncertainty of the measured values (Fig. 3). To assess how the different calibration scenarios influenced model results, the depth profiles of total OC, $\delta^{13}$C and $\Delta^{14}$C were plotted for every

calibration scenario (Fig. 3). To assess how each scenario influenced the range in optimal parameter values that resulted in behavioural models, the range in these parameter values for the different calibration scenarios was plotted (Fig. 4). All parameter values explored by the DE algorithm were plotted to confirm that the entire parameter space within the provided boundaries was explored (Fig. 4). Last, to assess correlations between parameters leading to behavioural models, correlation plots for these parameter values were developed using the Pearson correlation coefficient (Fig. 5).

## 2.5   Sensitivity analysis

### 2.5.1   Selection of calibration parameters

To assess the influence of the 11 parameters that were initially selected for optimization (Table S1), a sensitivity analysis was carried out using the PAWN method (Pianosi and Wagener, 2015). This is a density-based global sensitivity analysis that quantifies the model sensitivity related to uncertainties of input parameters based on the cumulative distribution function (CDF)

of the output distribution. The advantage of this method is that is does not assume that the variance of the model output is a measure for model uncertainty, making it more suitable to deal with e.g. multi-modal or skewed distributions than variance-based sensitivity analyses. The parameter sets to calculate the conditional and unconditional CDFs were obtained using the Matlab® version of SAFE toolbox (Pianosi et al., 2015), which was also used to post-process the results and calculate the sensitivity of the tested parameters using the Kolmogorov-Smirnov (KS) statistic. In addition to the parameters to be tested, a

dummy parameter with no influence on the model results was included in the sensitivity analysis. The KS statistic calculated for the dummy parameter was subtracted from the KS statistics of the model parameters before the results were analysed. We used 500 parameter sets to calculate the unconditional CDFs, 500 parameter sets to calculate the conditional CDFs and 50 conditioning values sampled from the one-dimensional space of each tested parameter, following recommendations by Pianosi and Wagener (2015). The parameter values were varied over the range that resulted in the 10 % best solutions in the first round

of model optimisation (see Sect. 2.3.2 and Table S1).

### 2.5.2   Sensitivity of parameters influencing the simulated $\delta^{13}$C depth profile

In a second sensitivity analysis, the sensitivity of the shape of the simulated $\delta^{13}$C depth profile to five model parameters was tested: (1) the $\delta^{13}$C value of OC inputs from aboveground biomass ($\delta^{13}C_{leaf}$), (2) the $\delta^{13}$C value of root OC inputs ($\delta^{13}C_{root}$), (3) the $\delta^{13}$C of rhizodeposit OC inputs ($\delta^{13}C_{exudates}$), (4) the fraction of microbial biomass derived from soil $CO_2$ ($\alpha$) and

(5) the change in fractionation against $^{13}$C by plants per unit change in atmospheric $CO_2$ concentration ($S$). The value of these parameters was varied over a range that results in a change in $\delta^{13}$C value of 1 ‰ (Table S4) to assure a uniform effect of the parameter ranges on the simulated depth profiles of $\delta^{13}$C, except for $\alpha$, which was varied over the range of values reported





in literature. The sensitivity of three characteristics of the simulated $\delta^{13}$C depth profiles was assessed: (1) the $\delta^{13}$C value in the top centimeter of the soil, the $\delta^{13}$C value at a depth of 0.40 m, and (3) the difference in $\delta^{13}$C between these two soil

layers. The sensitivity analysis was performed using the PAWN method (Sect. 2.5.1) to calculate the global sensitivity of these parameters. In addition, a local sensitivity analysis was performed by plotting depth profiles of $\delta^{13}$C for the range over which these parameters were varied during the global sensitivity analysis with the PAWN method.

## 3 Results

### 3.1 Simulation of depth profiles of OC, $\delta^{13}$C and $\Delta^{14}$C using the optimal parameters

The simulated amount of OC, $\delta^{13}$C and $\Delta^{14}$C in the litter layer closely align with measurements after parameter optimisation based on OC, $\delta^{13}$C and $\Delta^{14}$C data (Fig. S6). In addition, the temporal evolution of $\delta^{13}$C and $\Delta^{14}$C reflect changes in the value of these isotopes of atmospheric $CO_2$ over the past 150 years. Similarly for the soil, after parameters are optimised using measurements of depth profiles of OC, $\delta^{13}$C and $\Delta^{14}$C of the POC and MAOC pools, simulated depth profiles of these fractions closely reproduce measurements (Fig. 2). This indicates that the model captures differences in the amount of OC in POC and

MAOC, in addition to the residence time of OC in these pools. Concerning simulated depth profiles of $\delta^{13}CO_2$ and $\Delta^{14}CO_2$ along the soil profile (Fig. S8), the model simulated $\delta^{13}CO_2$ values of ca. 4 ‰ larger compared to available OC, and positive $\Delta^{14}CO_2$ values similar to the $\Delta^{14}$C available OC, in agreement with general observations (Trumbore, 2000; Cerling et al., 1991).

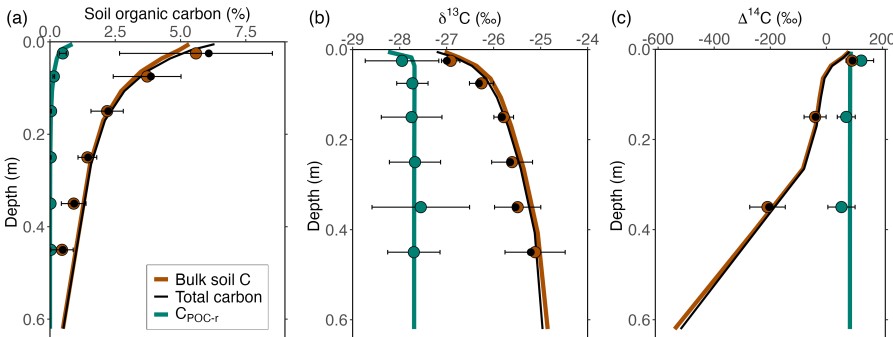

**Figure 2.** Simulated depth profiles of (A) OC(%), (B) $\delta^{13}$C(‰) and (C) $\Delta^{14}$C(‰) of the POC and bulk soil carbon pools (i.e., the sum of $C_{min-b}$, $C_{DOC-b}$ and $C_{mic-b}$), based on the calibration combining data on OC, $\delta^{13}$C and $\Delta^{14}$C for these pools. Circles indicate measured values for POC (green), mineral-associated carbon (brown) and total carbon (open circles) by Schrumpf et al. (2013). Error bars indicate the standard deviation on the measurements.



## 3.2 Simulation of OC depth profiles using different isotopic constraints

The results presented in Fig. 2 show an example of the model using the most optimal parameter set. However, a frequentist model optimisation that tunes model parameters to simulate the average measurements as closely as possible does not account for the fact that multiple parameter sets may result in a solution that is within measurement uncertainty (i.e., behavioural models). This was further explored by retaining the parameter sets that led to the 10 % best solutions obtained during the DE optimisation, as these were within the uncertainty of measurements (Fig. 3).

When the model is optimised using only data on the OC% of POC and MAOC (Fig. 3 A-C), simulated depth profiles of OC% show a close fit to measurements ($\bar{\epsilon} = 0.006$ [unitless]; $\bar{\epsilon}$ being the average weighted squared relative error for the POC and bulk soil C pools (i.e., MAOC), see Sect. 2.3.2), while simulated values of $\delta^{13}C$ along the depth profile are overestimated, most notably in the topsoil ($\bar{\epsilon} = 0.269$). Simulated $\Delta^{14}C$ values are underestimated for most simulations ($\bar{\epsilon} = 0.58$), with a large spread in simulated values, indicating that the turnover rate of OC along the soil profile is highly variable in this calibration scenario. For the second calibration scenario, the model was optimised using data on OC% and $\delta^{13}C$ of POC and MAOC (Fig. 3 D-F). Retained model results show a close fit between modelled and measured depth profiles of OC% ($\bar{\epsilon} = 0.036$) and $\delta^{13}C$ ($\bar{\epsilon} = 0.159$). Although simulations of topsoil $\Delta^{14}C$ show a closer fit with measurements compared to an optimization using OC% data only, the average $\Delta^{14}C$ values of SOC are overestimated below a depth of 0.2 m ($\bar{\epsilon} = 0.352$). This indicates that including $\delta^{13}C$ as a calibration constraint is not sufficient to correctly simulate $\Delta^{14}C$ values, and thus turnover rates, of subsoil OC. The

latter was only the case when data on $\Delta^{14}C$ was used to constrain model parameters (Fig. 3 G-L). Model optimisation using data on OC% and $\Delta^{14}C$, either with or without data on $\delta^{13}C$, resulted in a close fit between modeled and measured values of OC% ($\bar{\epsilon} = 0.049$ and $0.014$ resp.) and both isotopic ratios. It is noted that the simulated $\delta^{13}C$ depth profiles had a lower error when data on $\delta^{13}C$ was included as a calibration constraint ($\bar{\epsilon} = 0.181$) compared to when it was excluded ($\bar{\epsilon} = 0.241$).

The average error of OC% for the behavioral models was the lowest when only OC% data was used as a calibration constraint
($\bar{\epsilon} = 0.006$) and highest for the scenario using OC% data combined with $\delta^{13}C$ and $\Delta^{14}C$ ($\bar{\epsilon} = 0.049$). In contrast, the overall model error (calculated as the sum of the errors for the simulated depth profiles of OC%, $\delta^{13}C$ and $\Delta^{14}C$) was the lowest for the scenario constrained by all available data ($\bar{\epsilon} = 0.34$), while it was the highest for the optimization scenario using data on only OC% ($\bar{\epsilon} = 0.85$). This indicates that while the former optimization scenario does not result in the overall best fit for the simulated depth profiles of OC%, it results in the best overall model performance, given that processes such as the vertical
mixing of aboveground and belowground OC (as shown by the $\delta^{13}C$ values along the soil profile) and the turnover rate of OC (as shown by the $\Delta^{14}C$ values) are simulated more correctly.

## 3.3 Parameter equifinality

For the retained behavioural models, it was assessed how including different calibration constraints affected (1) the range and (2) the absolute values of the parameters (Fig. 4). In an ideal situation, the parameter values resulting in behavioural models
during a parameter optimization procedure (1) are correct in their absolute values and (2) do not show a large variation. To evaluate the first condition, it is assumed that the scenario in which parameter values are constrained using data on OC, $\delta^{13}C$




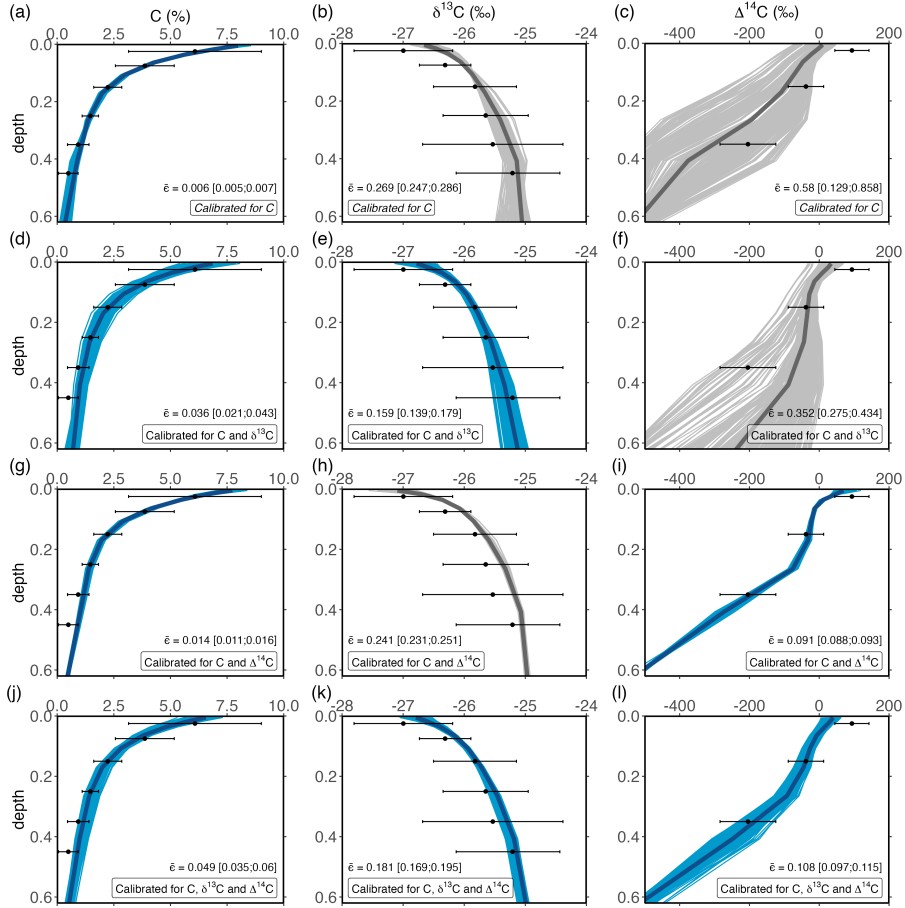

**Figure 3.** Optimised depth profiles of total OC, $\delta^{13}$C and $\Delta^{14}$C, obtained by optimizing the model using measurements of the POC and MAOC pools under different calibration scenarios: (1) optimisation using OC data only (A-C), (2) optimisation using OC and $\delta^{13}$C data (D-F), (3) optimisation using OC and $\Delta^{14}$C data (G-I) and (4) optimisation using C, $\delta^{13}$C and $\Delta^{14}$C data (J-L). These simulations show the 10 % best solution obtained using the DE algorithm. In each row, blue lines show depth profiles that were optimised, while grey lines show simulations of isotopes using the same optimised parameters. Dots and error bars show measured data by Schrumpf et al. (2013). The average error, calculated as a weighted average of the errors for POC and bulk soil OC (squared relative errors, see Sect. 2.3.2), is denoted by $\bar{\epsilon}$, the interquantile range is shown between squared brackets.

and $\Delta^{14}$C resulted in the most reliable parameter values, as this scenario led to the lowest average model error ($\bar{\epsilon} = 0.34$, Fig. 3) and most reliably simulated the turnover rate of SOC along the soil profile. Similarly, it was assumed that the parameter values of the scenario using data on OC only results in the least reliable parameter values ($\bar{\epsilon} = 0.85$). To evaluate the second criteria, 350 the interquartile range of the parameter values resulting in behavioural models was calculated (Fig. 4).

All model parameters were subject to equifinality for all calibration scenarios, i.e., there was always a range in parameter values that resulted in behavioural models (Fig. 4). Adding data on $\delta^{13}$C to the calibration constraints, in addition to data on



OC, improved only the value of the intensity of bioturbation ($D_b(0)$), i.e., resulting in values similar to the values obtained with the optimization scenario using data on OC, $\delta^{13}$C and $\Delta^{14}$C. As simulated depth profiles of $\delta^{13}$C are partly shaped by mixing

of aboveground and belowground OC, it is expected that adding information on the $\delta^{13}$C of OC better constrains parameters simulating this process. However, the optimal values of $D_b(0)$, after constraining with OC and $\delta^{13}$C, exhibited a substantial range. Adding data on $\Delta^{14}$C to the calibration constraints, in addition to data on OC, most notably improved the values of $k_{deprotect}(0)$, as is clear from the better prediction of the $\Delta^{14}$C value along the soil profile (Fig. 3 I). Other parameters had values different from the optimisation using C, $\delta^{13}$C and $\Delta^{14}$C, and/or had a substantial variation.

The most notable observation from these results is that for six out of the nine calibrated model parameters, including data on OC, $\delta^{13}$C and $\Delta^{14}$C did not result in substantially more constrained parameter values compared to when only data on OC is used (Fig. 4). These parameters regulate the simulated amounts of bio-available OC and DOC, two model pools that could not be explicitly constrained using available data. It thus seems that while adding data on $\delta^{13}$C and $\Delta^{14}$C better constrains parameters related to the turnover of the largest SOC pool (mineral-associated C), it does not help to constrain the size of other

model pools, which may compensate for each other to result in a correct amount of simulated total OC.

For all optimization scenarios, there were significant correlations between the optimized parameter values (as shown by the colored cells in Fig. 5 and Fig. S9). A first reason for the correlations between optimized parameter values is a consequence of the model structure, as parameter values can compensate inputs to and outputs from a model pool so its steady state size is similar. For example, when only data on OC was used to optimize model parameters (i.e., the optimization objective was only to

get a good fit between measured and modelled total OC%, irrespective of, for example, the turnover rate of OC), there is a strong correlation ($R^2$ = -0.69) between the rate of OC desorption from minerals ($k_{deprotect}(0)$) and the affinity of DOC to adsorption ($k_{m\_ads}$; note that lower values of $k_m$ imply a higher affinity). This is to be expected, as low and high rates, respectively, of both C inputs and outputs to mineral-associated OC, will lead to a similar size of this pool, although with respective slow and fast turnover rates. In contrast, when the turnover of the mineral-associated OC pool is included as a calibration criterion

(through its $\Delta^{14}$C value), this correlation is absent, as only a narrow range in desorption rates ($k_{deprotect}(0)$) result in the correct turnover rate of this pool (Fig. 4 (I)).

A second reason for such correlations is related to the formulation of the mathematical equations. For example, parameters in the numerator and denominator of an equation may compensate for each other. This is clear from the scenario including most optimization data (Fig. 5 (B)), where there is strong correlation between $V_{max\_ads}$ and $K_{m\_ads}$, which occur in the numerator

and denominator, respectively, of the equation representing the rate of DOC adsorption on minerals. While such correlations are generally unwanted (they are an expression of equifinality) and complicate the optimization procedure, they reflect the ability of the optimization algorithm to find parameter values that lead to a narrow range of adsorption rates, resulting in the correct simulation of the turnover time of the mineral-associated OC pool.

### 3.4 Sensitivity of parameters affecting simulated $\delta^{13}$C depth profiles

The global sensitivity analysis of parameters affecting the $\delta^{13}$C value of both topsoil and subsoil OC (Fig. 6, A - C) showed that the $\delta^{13}$C value of topsoil OC was most influenced by the $\delta^{13}$C value of leaves, while the other tested parameters had a



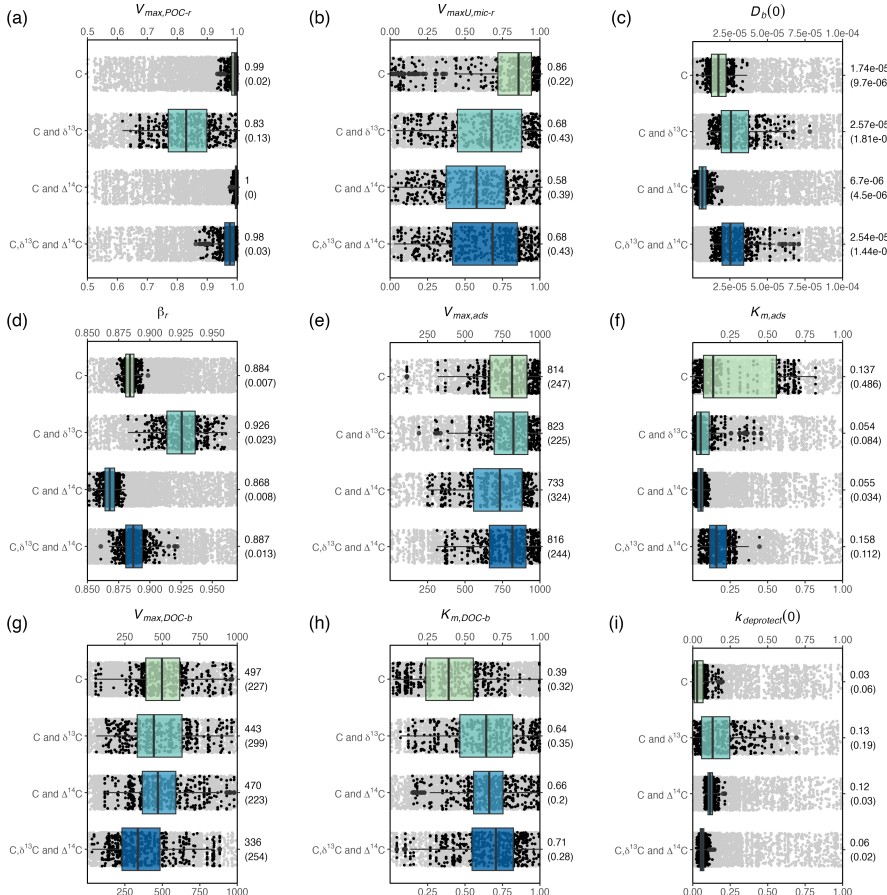

**Figure 4.** Ranges in parameter values that resulted in the retained model simulations in Fig. 3. The ranges are shown for all optimised parameters, grouped per calibration scenario. Black dots show the retained parameter values, while boxplots show the quantiles of these optimal parameter values. Grey dots show all parameter values tested by the DE algorithm in the range of the retained values. The median and interquartile range (between brackets) of the retained values are shown on the right side of the graphs.

limited effect. The subsoil (0.40 m depth) $\delta^{13}$C was influenced most by the $\delta^{13}$C of roots and the effect of atmospheric $CO_2$ concentration on isotopic fractionation against $\delta^{13}CO_2$ during photosynthesis. The influence of the $\delta^{13}$C value of leaves on the subsoil $\delta^{13}$C was negligible, indicating that the incorporation of aboveground biomass into the soil profile was limited to the uppermost soil layers. The change in the $\delta^{13}$C value along the soil profile ($\Delta^{13}$C topsoil - subsoil) was most sensitive to the $\delta^{13}$C leaves, the $\delta^{13}$C of roots and the effect of atmospheric $CO_2$ concentration on isotopic fractionation against $\delta^{13}CO_2$ during photosynthesis. The local sensitivity analysis (Fig. 6, D - H) confirmed these results, showing that the factors having the largest effect on absolute values of $\delta^{13}$C along the soil profile were the $\delta^{13}$C values of leaves and roots, and the effect of atmospheric $CO_2$ concentration on isotopic fractionation against $\delta^{13}CO_2$ during photosynthesis.





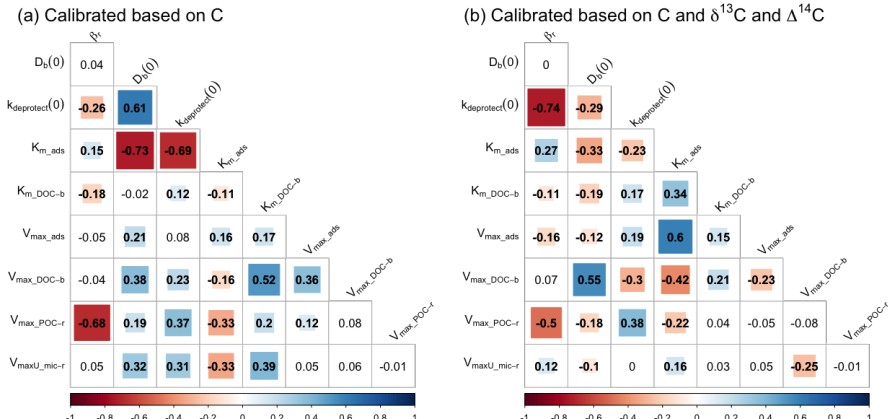

**Figure 5.** Correlation between the optimised parameters for the calibration scenarios using data on (A) OC only and (B) OC, $\delta^{13}$C and $\Delta^{14}$C (B). Numbers are the correlation coefficients, while colors are shown for parameter combinations with a significant correlation ($p < 0.05$). Correlation plots for the other calibration scenarios are shown in Fig. S9

.

## 4   Discussion

### 4.1   Simulation of $\delta^{13}$C depth profiles in SOILcarb: mechanisms and challenges

During the past decades, multiple mechanisms have been put forward to explain the generally-observed increase in the $\delta^{13}$C value of SOC with depth in temperate ecosystems. Here, it is assessed to which extent these mechanisms are reflected in the model outcomes. Three main mechanisms that have been proposed are simulated by SOILcarb. The first mechanism concerns inputs of $^{13}$C-depleted aboveground litter at the soil surface and vertical mixing with $^{13}$C-enriched belowground inputs along the soil profile (e.g. Wynn et al., 2006; Jagercikova et al., 2017). The sensitivity analysis showed that this mechanism plays an important role in shaping the depth profile of the $\delta^{13}$C value of SOC at the studied site (Fig. 6D). This effect played a role down to a depth of ca. 0.3 m, as shown by a model simulation in which the mixing of above- and belowground vegetation was the only mechanism affecting the $\delta^{13}$C depth profile (Fig. S10). The second mechanism concerns temporal variations in the $\delta^{13}$C value of vegetation, and thus C inputs to the soil (e.g. Paul et al., 2019; Wynn et al., 2006). In SOILcarb, this effect has been partitioned into (1) temporal changes in the $\delta^{13}$C value of atmospheric $CO_2$ (Keeling, 1979) and (2) the effect of atmospheric $CO_2$ concentration on the discrimination against $^{13}CO_2$ during photosynthesis (i.e., a higher atmospheric $CO_2$ concentration leads to more intense fractionation against $^{13}CO_2$ by plants, and thus lower $\delta^{13}$C values (Schubert and Jahren, 2012)). Additional simulations with SOILcarb show that when only the first mechanism is considered, the simulated $\delta^{13}$C of SOC increases with ca. 1 ‰ with depth (Fig. S11). While this process thus contributes substantially to the observed increase in $\delta^{13}$C with depth, including the effect of atmospheric $CO_2$ concentration on the fractionation against $\delta^{13}CO_2$ during plant photosynthesis was necessary to simulate the measured increase in $\delta^{13}$C with soil depth of ca. 2 ‰ (Fig. 2B). The last mechanisms is heterotrophic



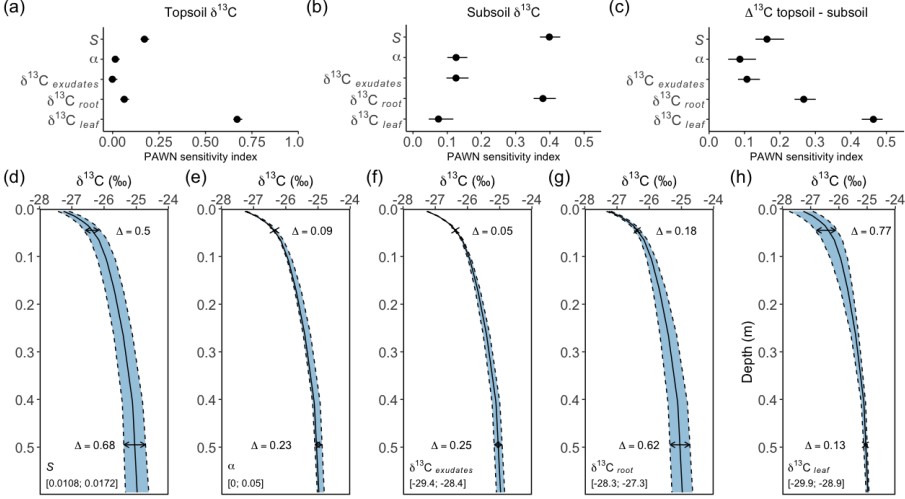

**Figure 6.** Sensitivity of simulated $\delta^{13}C$ depth profiles to five model parameters: (1) the change in fractionation against $^{13}C$ by plants per unit change in atmospheric $CO_2$ concentration ($S$), (2) the fraction of microbial biomass carbon derived from $CO_2$ assimilation ($\alpha$), (3) the $\delta^{13}C$ of rhizodeposit OC inputs ($\delta^{13}C_{exudates}$), (4) the $\delta^{13}C$ value of root OC inputs ($\delta^{13}C_{root}$) and (5) the $\delta^{13}C$ value of leaf OC inputs ($\delta^{13}C_{leaf}$). The sensitivity of these parameters is calculated for topsoil $\delta^{13}C$ (0.01 m depth), subsoil $\delta^{13}C$ (0.40 m depth) and the difference in $\delta^{13}C$ between these layers. The top row (A - C) shows the results for the global sensitivity analysis, while the bottom row (D - H) shows the results for the local sensitivity analysis. The $\Delta$ values in the lower row (D - H) indicate the maximum difference in the simulated $\delta^{13}C$ values for the topsoil and subsoil, while the ranges in the lower left corners of these graphs show the range over which the respective parameter values were varied.

$CO_2$ assimilation by soil microbes (e.g. Šantrůčková et al., 2018; Nel and Cramer, 2019). The sensitivity analysis showed that this was the mechanism with the lowest impact on the difference in $\delta^{13}C$ between the topsoil and the subsoil, with the effect
on the range in subsoil $\delta^{13}C$ being 0.23 ‰ when the value of $\alpha$ varies over the range reported in the literature (Fig. 6E). Our model simulations thus suggest that, in contrast to proposals of this mechanism being important based on empirical studies, the potential effect is limited. At the study site, this is caused by the limited amount of $CO_2$ that is assimilated by soil microbes (1.1 % of total microbial biomass; Akinyede et al. (2020)) and the limited difference between the $\delta^{13}C$ value of SOC and soil $CO_2$ of 4.4 ‰ (Cerling et al., 1991).
While numerical simulations of depth profiles of $\delta^{13}C$ help to quantify the importance of different mechanisms shaping the vertical profile, these simulations are prone to uncertainties. For example, the absolute values of the $\delta^{13}C$ of SOC depend on the $\delta^{13}C$ value of vegetation (Fig. 6F - H). In the present study, we relied on measurements made at the study site to obtain this information, but these measurements are often not available. Estimating the $\delta^{13}C$ of vegetation based on literature values when measured data is not available is unlikely to be reliable, as $\delta^{13}C$ values of C3 vegetation vary over a large range (between
ca. -23 - -32 ‰) depending on, for example, precipitation (Kohn, 2010) and vegetation type (Martinelli et al., 2021). Also the $\delta^{13}C$ value of different plant organs varies considerably (Bowling et al., 2008). Most notably, estimating the $\delta^{13}C$ value of root





exudates is challenging. Thus, there is considerable uncertainty on estimates of the $\delta^{13}$C value of different sources of OC inputs to the soil. Therefore, model users should be aware of large uncertainties in simulated absolute values of the $\delta^{13}$C of SOC when measured values are not available, and thus balance the benefits of simulating $\delta^{13}$C versus the increased uncertainty.

## 4.2 Overparameterisation and equifinality in soil biogeochemical models

Overparameterisation, which arises when a numerical model has too many parameters compared to the data available to constrain parameter values, is a general problem in biogeochemical modeling (Braakhekke et al., 2013; Luo et al., 2016, 2017). This is also illustrated by our simulations. Using only data on total C concentration of the POC and MAOC pools to constrain parameter values resulted in many parameter combinations that led to behavioural models for total C, i.e. with a close fit to
observations. However, simulated $\Delta^{14}$C values within the soil profile were generally underestimated, while $\delta^{13}$C values were slightly overestimated (Fig. 3A-C). This does not confirm our second hypothesis, which anticipated an overestimation of the turnover rate of SOC. Nevertheless, this shows that, if only total SOC stocks are used as the calibration criterion, turnover times of SOC are simulated incorrectly, despite a correct simulation of the total SOC inventory. This is important, as a correct simulation of the turnover time of SOC is crucial to make reliable projections of changes in the global C cycle for the coming
decades (He et al., 2016; Wang et al., 2019). Similar conclusions were drawn at the plot scale by Braakhekke et al. (2014), who found that the turnover rate for the slowest SOC pool in their model was substantially overestimated without $\Delta^{14}$C data as a constraint on parameter values. Furthermore, for simulations of the turnover rate of SOC at the global scale, models optimised without data on $\Delta^{14}$C resulted in a substantial overestimation of the turnover rate of SOC (He et al., 2016). It is thus clear that, without data on the age of SOC as a parameter constraint during calibration, the turnover rate of SOC, especially in the subsoil,
is unlikely to be simulated correctly.

Soil biogeochemical models not only suffer from overparameterisation, but also from parameter equifinality, i.e. the phenomenon that multiple parameter sets lead to model results that cannot readily be rejected (Marschmann et al., 2019; Sierra et al., 2015; Tang and Zhuang, 2008). In line with our first hypothesis, a model constrained by data on only SOC stocks was characterised by substantial equifinality (Fig. 3 A-C). However, contrary to our third hypothesis, including data on the $\delta^{13}$C
and/or $\Delta^{14}$C values of SOC to constrain parameter values during calibration did not substantially reduce the range in most parameter values leading to behavioural models, as shown by the interquartile distances in Fig. 4. Two exceptions were the range in rates of deprotection of OC ($k_{deprotect}(0)$) and affinity of DOC for adsorption ($k_{m,ads}$), which were substantially reduced when data on $\delta^{13}$C and $\Delta^{14}$C were included during optimization.

In line with previous studies, we found that the parameters of the Michaelis-Menten equation ($V_{max}$ in the numerator and $K_m$
in the denominator) were subject to substantial equifinality (Sierra et al., 2015; Marschmann et al., 2019). The wide use of this equation in microbially-driven soil biogeochemical models thus suggests that equifinality of these parameters is common, as information on both the maximum rate (represented by $V_{max}$) and the rate-limiting property (represented by $K_m$) is generally not available. Other parameters of SOILcarb subject to equifinality are representing processes that can compensate for each other to result in a similar total pool size. For example, as shown by the positive correlation between the rate of bioturbation



$(D_b(0))$ and the rate of C uptake by microbes in the bulk soil ($V_{max,DOC-b}$): the more OC in the bulk soil diffuses downwards, the faster microbes need to process it to simulate the measured OC content.

### 4.3  Ways forward to identify and reduce equifinality in microbially-driven SOC models

Sierra et al. (2015) show that equifinality is likely to be an issue in microbially-driven SOC models. These authors used the identifiability analysis by Brun et al. (2001) to show that for a relatively simple non-linear microbial model, only 2 or 3
parameters could be uniquely identified using calibration data on soil respiration and $\Delta^{14}$C values of bulk soil and respired $CO_2$. Similarly, Marschmann et al. (2019) studied five microbially-driven SOC models of varying complexity, and found substantial equifinality in every model, including a simple 2-pool microbial models. From previous studies and the results presented here, it seems that equifinality in soil biogeochemical models can only be partly reduced by including more generally available data, while a reduction in complexity might be needed to fully resolve this issue.

A consequence of equifinality is that it undermines confidence in projected changes in the SOC stock due to environmental changes, as behavioural models can make similar projections for the near future, but greatly diverge on a decadal timescale (Luo et al., 2016, 2017). Therefore, identifying and reducing equifinality in soil biogeochemical models is an important prerequisite to increase confidence in the predictions by such models.

One way forward to better constrain parameters in microbially-driven SOC models is to include additional data during the
parameter calibration process. As show in the present and previous studies (He et al., 2016; Wang et al., 2019), the residence time of SOC along the soil profile can be better constrained by including data on $\Delta^{14}$C during calibration. Reducing the range in acceptable parameter values related to soil microbial dynamics is, however, more challenging, as this data is often lacking, especially for the subsoil or over large spatial scales. Therefore, it is likely that parameters related to soil microbial dynamics in soil biogeochemical models will have to be optimised until more data become available, or fixed at values derived from
measurements.

Equifinality also implies that it is unlikely that the development of even more complex models will immediately pay off in terms of improved accuracy in predictions. Defining the optimal model structure for simulation and prediction, given the data that are available, is therefore as important as further increasing our process understanding. Multiple methods are available to identify parameter equifinality in environmental models, including the GLUE methodology (Beven and Binley, 1992),
Bayesian methods (Vrugt, 2016), the parameter identifiability method from Brun et al. (2001), the Manifold Boundary Approximation Method (Marschmann et al., 2019) and methods to assess local structural parameter identifiability (Stigter et al., 2017), among others (Miao et al., 2011). Many of these methods are easily accessible to researchers in the form of packages in *R* and other software environments. This should enable modellers to identify this phenomenon in their models and thus reduce model complexity when appropriate.

A last way forward to better constrain model parameters is the construction of integrated databases that bring together data on multiple aspect of the SOM cycle (e.g. OC fractionation, stable and radioisotopes, mineralogy, microbial characteristics, environmental drivers, etc.) (Sierra et al., 2015). While in the recent past several efforts have been made to construct global databases with data related to SOC cycling (e.g. ISRAD ((Lawrence et al., 2020)), WOSIS (Batjes et al., 2020), SoDaH (Wieder

et al., 2021), LUCAS (Orgiazzi et al., 2018), among others), the use of these databases to identify and reduce equifinality in soil
biogeochemical models has been, surprisingly, very limited. Thus, this is a low-hanging fruit that would significantly increase
our confidence in projections of the soil carbon - climate feedback for the coming decades.

## 5 Conclusions

In this study, a new mechanistic, depth-explicit SOC model (SOILcarb) was presented and used to assess the potential to
decrease parameter equifinality by including data on $\delta^{13}$C and $\Delta^{14}$C data of two soil fractions (POC and MAOC) as constraints
on parameter values during model optimisation. Our results show that while the optimized model was able to simulate depth
profiles of total OC, $\delta^{13}$C and $\Delta^{14}$C in line with measurements, all optimised model parameters were prone to equifinality.
Including $\delta^{13}$C data, in addition to total OC, did little to improve simulations of the turnover rate of SOC or limit parameter
equifinality. Adding $\Delta^{14}$C data as a calibration constraint, in contrast, allowed the correct simulation of the turnover rate of
SOC, while only substantially reducing equifinality for the parameter regulating desorption rate of OC from minerals. Our
results show that more data is needed to reliably constrain parameter values of microbially-driven models. As these data are
generally not available at larger spatial scales, it is unlikely that including more complexity in soil biogeochemical models
will improve simulations in the near future, while more emphasis should be put on finding a better balance between model
complexity and available data. This is an important prerequisite to increase confidence in projections of the soil carbon -
climate feedback in a world subject to climatic change.

*Code availability.* The R codes of SOILcarb have been submitted together with this manuscript for review purposes. Upon acceptance of the
manuscript, the codes will be made publicly available on the GitHub page of the first author.

*Author contributions.* MVdB developed the model codes with inputs from all co-authors, and took the lead in writing the manuscript. All
co-authors contributed to writing the final submitted manuscript.

*Competing interests.* The authors declare that they have no conflict of interest.

*Acknowledgements.* This work was partly supported by the Swiss National Science Foundation (SNSF, Ambizione grant number PZ00P2_193617
/ 1, granted to MVdB).



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
