# Peer review of "A microbially-driven and depth-explicit soil organic carbon model constrained by carbon isotopes to reduce parameter equifinality"

_EGUsphere, 2024_

## Author Comment (AC1)

Feedback from the reviewer is written in italic face, while our responses are written in normal font in green.

**Reviewer 1**

*The article is devoted to the urgent challenges of microbially-driven SOC models. The authors created numeric mirobially-driven and depth explicite model of SOC dynamics using R code. They used different scenarios based on isotopic constraints to optimize model parameters, assessed parameter equifinality and checked model sensitivity related to uncertainties of input parameters. The authors revealed that numerical simulations of depth profiles of δ13C are prone to uncertainties connected with data availability, wide range of δ13C values of C3 vegetation, challenges in estimation of δ13C value of root exudates. One more important findings is that despite the including data on the δ13C and/or Δ14C values of SOC to constrain parameter values during calibration did not substantially reduce equifinality of the most parameters, Δ14C data needs to be incorporated to the model calibration for correct simulation of the turnover time of SOC (models without this data substantially overestimated SOC turnover rate). To include additional data during the parameter calibration process is one way forward to improve microbially-driven SOC models. However, they advise avoiding overparametrisation which lead to behavioural models. Defining the optimal structure of soil biogeochemical models and finding a balance between model complexity and available data is an important prerequisite to increase confidence in global projections of the soil carbon - climate feedback. Also, the authors suggests to create and use global databases with data related to SOC cycling to better constrain model parameters.*

*This article spotlights the biogeochemical modeling challenges and can help to improve simulation of SOC dynamics. But I have several questions concerning modeling methodology and I would appreciate if the authors explain some of the details. I have also added suggestions to improve the quality of the illustrations in the article and in the Supplementary information file. See attached files with comments.*

We thank the reviewer for taking the time to read our manuscript and for providing detailed and constructive feedback. Please find our responses to the feedback below.

Specific comments

*Title: M.b. "to reduce parameter equifinality" or "to reduce model equifinlity"*

We thank the reviewer for this suggestion, we changed the title to "[...] parameter equifinality", so it's clear to the reader that it's this aspect of equifinality the manuscript is about.

*Line 97: See the comments and questions in the Supplementary Information file*

Thanks for having a detailed look at the supplement, we address this feedback below.

*Line 152: As I saw in Supplementary Information user can choose any other year.*

This should indeed be the last simulated year, this has been changed in the supplement: "*[...] $\delta^{13}C$ value of atmospheric $CO_2$ for the last simulation year*:"

*Line 157-158: Please explain why this is so*

As explained in a handbook on radiocarbon in the environment by Schuur et al. (2016, section 3.3.2, p 54; doi.org/10.1007/978-3-319-25643-6), the fractionation against $^{14}CO_2$ is roughly twice of that against $^{13}CO_2$, compared to $^{12}CO_2$, because the mass difference between $^{12}CO_2$ and $^{14}CO_2$ is twice than that between $^{12}CO_2$ and $^{13}CO_2$ (the atomic mass of $^{12}CO_2$, $^{13}CO_2$ and $^{14}CO_2$ is approximately 44.01 g/mol, 45.01 g/mol and 46.01 g/mol).

This has been clarified in the manuscript: "*It was assumed that during photosynthesis, the fractionation against $^{14}CO_2$ is twice that of against $^{13}CO_2$, as the mass difference between $^{14}CO_2$ and $^{12}CO_2$ is twice than that between $^{13}CO_2$ and $^{12}CO_2$ (Schuur et al., 2016).*"

*Line 189: But if you excluded occluded light fraction it is not total OC...*

That's correct. To make this clear to the reader, and avoid having to indicate this every time total SOC is mentioned, we now added the following: "*As SOILcarb does not simulate aggregate dynamics, the total amount of measured OC was reduced by the amount of OC in the occluded light fraction, which constituted 8.4% of total SOC down to 60 cm. Therefore, when referring to total SOC in this manuscript, we refer to the sum of POC and MAOC.*"

*Line 209: Do you mean you analysed root exudates?*

No, we did not do this. Since we had no information on the $\delta^{13}$C value of root exudates, or data from other studies to fall back to, we ran the model with a range of reasonable values of the $\delta^{13}$C of root exudates, and choose the value that resulted in the $\delta^{13}$C value of SOC closest replicating the measurements. This resulted in the value for the $\delta^{13}$C of root exudates used in the simulations of -28.9 ‰.

To clarify this for the reader, we changed these sentences to: "*As measurements of the $\delta^{13}$C value of root exudates were not available, a range of reasonable values was tested and the resulting $\delta^{13}$C values of SOC and MAOC depth profiles were compared to measured values. The tested $\delta^{13}$C of root exudates that resulted in the closest fit of measured and modelled depth profiles of $\delta^{13}$C was -28.9 ‰, which was used for all subsequent simulations.*" We hope that this clarifies this strategy for the reader.

*Line 210: Please explain in more details. [Resulting in an optimal $\delta^{13}$C value for root exudates of -28.9 per mil]*

Please see the response to the previous question.

*Line 223: Why? [33 % as DOC, 66 % as POC and 1 % as microbial C]*

The choice for this distribution was made based on our expectations of reasonable values. As we did not have any data on this, and the focus of the study is on soil OC, our aim was to calibrate the litter compartment of the model to result in reasonable values, which could be used as C inputs to the soil.

To make this more clear for the reader, we changed this to: "*No information on the distribution of the total amount of litter C between the simulated model pools ($C_{POC-l}$, $C_{DOC-l}$ and $C_{mic-l}$) was present. As the focus of the present study is on OC dynamics in the soil, the amount of measured OC in the litter layer was assumed to be distributed as follows: 33 % as DOC, 66 % as POC and 1 % as microbial C. We note that these portions were not based on data, but on our best estimates of a reasonable distribution of OC in the litter layer of a temperate forest.*"

*Figure 2: May be black? [(open circles)]*

This should indeed be "black circles", thanks for noticing this.

*Figure 2: Please give the error bars in the color of circles to which they are corresponded.*

Thanks for this suggestion, this has been changed.

*Line 334: behavioural*

Thanks for noticing this mistake, this has been corrected.

*Line 360-362: For which three parameters did the inclusion of OC, 13C, and 14C data result in a significant increase in constraints? Km ads, Km DOC-b, and what is third one?*

Also $K_{deprotect}(0)$. This is indeed not easy to see in Figure 4, but the difference in values, and better constraints, has a large effect on the simulated $\Delta^{14}C$ values (Figure 3). To make this clear to the reader, we now explicitly mention these 3 parameter names in the sentence: *"The most notable observation from these results is that for six out of the nine calibrated model parameters (all except $_{Km,ads}$, $K_{m,DOC-b}$ and $k_{deprotect}(0)$) [...]".*

*Line 384: Why did you miss in this section absorpation as a mechanism influencing 13C depth profile?*

The reason for not including absorption in the sensitivity analysis is that in the model, $^{12}C$ and $^{13}C$ are absorbed on minerals at the same rate. As a result, the $\delta^{13}C$ value of mineral-associated OC is determined by the $\delta^{13}C$ value of DOC and microbial residues, which are in their turn influenced by the factors included in the sensitivity analysis (such as the $\delta^{13}C$ value of roots, exudates and aboveground biomass). For this reason, absorption as a process was not included in the sensitivity analysis.

As this may not be clear to the reader, we included the following sentence in section 2.5.2 (which describes the methods of the sensitivity analysis): *"We note that the process of absorption was not included in this sensitivity analysis, as there is no preferential absorption of $^{12}C$, $^{13}C$ or $^{14}C$ on minerals in the model."*

*Figure 4: Please show all labels in black. And it would be better you enlarge the labels*

Thanks for this suggestion, we gave all labels a black color. It would indeed have been better to increase the size of the labels, but in that case the labels on the x-axis overlap. Therefore, we have to keep the original size of the labels.

*Line 386-387: It seems like collinearity can excist between this parameter and 13C of leaves. Have you check correlation between these parameters?*

That is correct, the parameter $S$ (representing the effect of the atmospheric $CO_2$ concentration on the discrimination against $^{13}CO_2$ during photosynthesis) directly affects the $\delta^{13}C$ value of leaves, so their values will be correlated (the larger the value of S, the lower the $\delta^{13}C$ value). As the parameter of $S$ is fixed and the effect is unidirectional ($S$ affects the $\delta^{13}C$ value of biomass), this should not be a problem for, e.g., equifinality.

As this effect is generally not included in models simulating the $\delta^{13}C$ value of SOC, we included this separately in the sensitivity analysis (on top of changing the $\delta^{13}C$ value of leaves and roots), to make the magnitude of this effect clear to the reader.

*Figure 5: Please enlarge these figures (labels are illegible).*

Thanks for this suggestion, we enlarged the labels.

*Line 406-408: As for me this effect is reflected in 13C composition of leafs. So, as I have already mention above thes parameter can be collinear.*

As mentioned above, both parameters (the simulated $\delta^{13}C$ value and the value of the parameter $S$) are probably collinear. The two processes mentioned in these lines are, however, different, and affect the $\delta^{13}C$ value of vegetation over different timescales:

- The first parameter (changes in the $\delta^{13}C$ value of vegetation due the temporal changes in the $\delta^{13}C$ value of atmospheric $CO_2$, mainly due to fossil fuel burning) affected the $\delta^{13}C$ value of vegetation mainly since the 1950s
- The second parameter (changes in the $\delta^{13}C$ value of vegetation due to differences in the concentration of atmospheric $CO_2$) affects the $\delta^{13}C$ of vegetation over 10,000s of years, as the $CO_2$ concentration of the atmosphere has not been consistently stable over this period.

Therefore, we would like to keep this differentiation in the manuscript.

*Line 502-504: What about combination of OC, 13C and 14C?*

That was indeed missing here, thanks for pointing this out. We now added: "*Adding a combination of $\delta^{13}C$ and $\Delta^{14}C$ data improved the simulation of the $\delta^{13}C$ value in the topsoil, and the rate of sorption and desorption of OC on minerals along the soil profile, and thus the turnover rate of SOC along the soil profile.*"

**Feedback on supplementary information**

*Line 44: plant-derived OC adsorbed onto soil minerals? Unclear how*

The reason for this simulated flux is to account for the fact that a significant portion of mineral-associated OC has a plant origin (opposed to microbial origin), as shown by recent review articles (Wang et al., 2021 (https://doi.org/10.1016/j.soilbio.2021.108422, 2021); Angst et al., 2021 (https://doi.org/10.1016/j.soilbio.2021.108189, 2021)). Therefore, the simulated bioavailable C pool (consisting of root exudates and depolymerised POC) can be directly associated with soil minerals.

*Line 64: Vmax_POC_l?*

That's correct, thanks for pointing this out.

*Line 64: Vmax_DOC_l?*

That's correct, thanks for pointing this out.

*Line 106: This is unclear. You have told about transferring of the litter POC pool to the rhizosphere POC pool before the formula, but after that you tell that the litter POC pool is trasferred to the soil POC pool.*

You are right, "soil POC pool" has been changed to the "rhizosphere POC pool"

*Line 174: Is it right? May be (1-fsol)?*

Yes, it is correctly formulated. The soluble part of microbial necromass (e.g. the cytoplasm) is transferred to the bio-available C pool, as the assumption in the model is

that this C can be taken up readily be microbes. The non-soluble part (e.g. the cell wall) is transferred to the DOC pool, which needs to be depolymerized before being taken up by microbes (although depolymerisation and uptake are simulated as a single-step process in the bulk soil).

*Line 175: Is it right? May be fsol?*

See the previous response

*Line 195: double backets*

Thanks for pointing this out

*Line 331-336: It is better to shift this part to previous previous paragraph (325)*

Thanks for this suggestion. We followed this, and made some other small changes to this paragraph.

*Line 363-364: Have you checked if the "diff_fixed" values are different for other years? If so, how much do the values change?*

The values of *diff_fixed* is the same for every simulation year, it's a measure for the difference in $\delta^{13}C$ between atmospheric $CO_2$ and plant biomass. The variable part is added to account for additional fractionation against $^{13}CO_2$ due to differences in the concentration of $CO_2$ in the atmosphere for every simulated year.

*Line 376: Why have you taken double discrimination?*

Please see our response to the same question above.

*Line 388: I think that you'd better give the value you choose in this section (not only in the Table S3). And for "diff13C_leaf-exudates" too.*

Thanks for this suggestion, we added these values to the text in this section

*Figure S5: (7)*

Thanks for noticing this, this has been changed.

*Figure S9: Please enlarge these figures (labels are illegible).*

The size of the labels has been enlarged.

*Figure S10: Please label each figure (A, B, C) in the figure caption.*

This has been added.

*Figure S11: Please label each figure (A, B, C) in the figure caption.*

This has been added.

*Figure S11: What about δ13C of atmospheric CO2 ?*

The temporal variation in the $\delta^{13}$C of atmospheric $CO_2$ is what drives the temporal changes in the $\delta^{13}$C of vegetation (the process being simulated). To make this clear to the reader, this has been added to the caption: "[*(ii) temporal changes in the $\delta^{13}$C of vegetation (due to temporal variations in the $\delta^{13}$C value of atmospheric $CO_2$)*] ".

*Table S6: Can you add references for fixed values (where it is possible)?*

Where possible, references for the fixed parameter values are provided in the detailed model description in the supplement. Other fixed parameter values where fixed to realistic values, to not further complicate the model calibration process. To make the reader clear where references can be found, the following sentence was added to the caption: "*Where possible, references to fixed values are provided in the detailed model description above.*".

*Table S6: "?"*

This parameter is used to create simulated soil layers which increase in thickness with depth. This is explained in section 1.1, and is used in equation (1).

*Table S6: You missed parameter "surf" in this section.*

This is not a parameter but is used as a state variable in the model. This should thus be added to table S5 (thanks for noticing this). We did so, and added a note stating that this parameter is calculated by subtracting the actual amount of MAOC from the total potential amount of MAOC.

*Table S6: I did not see "m" in this section. And the discription of "m" is the same with "α".*

*The parameter m is used in equation (43). To make the difference between m and alpha more clear to the reader, their description has been changed. For m this is now "Coefficient to calculate the effective gas diffusivity", and for alpha "Coefficient to calculate gas diffusivity of $CO_2$ in free air".*

---

## Author Comment (AC2)

Feedback from the reviewer is written in italic face, while our responses are written in normal font in green.

**Reviewer 2**

We thank the reviewer for taking the time to read our manuscript and for providing constructive feedback. Please find our responses to the feedback below.

*Manuscript Title*

*A microbially-driven and depth-explicit soil organic carbon model constrained by carbon isotopes to reduce equifinality*

*Recommendation*

*This paper presents a novel SOC model (SOIL carb), designed to mitigate equifinality by integrating $\delta^{13}C$ and $\Delta^{14}C$ values of soil organic carbon (SOC). Calibration solely based on SOC stock data results in imprecise estimations of subsoil organic carbon (OC) residence times. The inclusion of $\delta^{13}C$ has a minimal effect, whereas the incorporation of $\Delta^{14}C$ accurately captures the SOC turnover rate but only partially alleviates equifinality for certain parameters. Given that all parameters are susceptible to equifinality, additional data is required to establish reliable constraints. Achieving an optimal balance between model complexity and data availability is crucial for accurately predicting soil carbon-climate feedback mechanisms.*

*The article's topic selection is significant and demonstrates robust logic and academic rigor. Nevertheless, certain sections necessitate further refinement. The subsequent revision recommendations are outlined below*

*Major revisions*

*Introduction*

1. *Further explanation can be provided on why accurately predicting the reserves and dynamics of Soil Organic Carbon (SOC) is crucial for combating climate change. Additionally, pointing out the problems caused by inaccurate SOC models, it can enhance readers' comprehension of the urgency and significance of this research endeavor.*

Thanks for this suggestion, this was indeed missing in the introduction. At the end of the first paragraph of the introduction (L26-28 in the original manuscript), we added: "*A correct representation of the rate of OC cycling along the soil profile in biogeochemical models is necessary to make accurate predictions about climate – soil carbon feedbacks. When these rates are overestimated, the simulated size of the SOC stock*

*will adapt too fast to changes in OC inputs. This leads to an underestimation of the time it takes for soils to increase their OC storage due to increases in, for example, net primary productivity or OC inputs in agroecosystems (He et al., 2016; Wang et al., 2019).".*

To make the potential impact of equifinality in SOC models clear, we added to following to L60 (in the original manuscript): "*In the case of SOC models, models characterised by equifinality are often able to make correct predictions of current SOC stocks, although these stocks can be predicted by different distributions of SOC over the simulated model pools (Braakhekke et al., 2013). The problems (and uncertainty) arise when different behavioural models are used to make predictions of SOC stocks based on changing environmental conditions or OC inputs. In this case, behavioural models starting from an identical initial SOC stock can produce a wide range in predicted values, from which it is generally not possible to identify the correct model (and parameter set) (Luo et al., 2016, 2017).".*

2. *It is suggested that a brief discussion be included at the end of the introduction regarding the potential impacts of this study on soil carbon cycling, climate change prediction, and land management practices, enhancing the practicality and relevance of the research.*

Thanks for this suggestion. We now end the introduction with "*As equifinality in SOC models has received only limited research attention, increasing awareness of, and solving, this problem will increase confidence in simulations of the role soils can play in climate change mitigation or increasing SOC stocks to improve soil health in agroecosystems.".*

**Materials and Methods**

1. *In Sensitivity Analysis: The authors should provide more details on the parameter sensitivity analysis, particularly for those parameters that have the greatest impact on the model's output.*

*The description of the methods for the sensitivity analysis (section 2.5.1) has been updated to make this more clear (see below). The results of this sensitivity analysis are briefly presented in section 2.3.2, and figures showing the results are provided in the supplementary information (Fig. S5). We hope this is sufficient for the reader to understand these results.*

**Results**

1. *At the beginning of each results section paragraph, the key findings of this study can be highlighted using concise and clear language, enabling readers to quickly grasp the main outcomes of the research.*

Thanks for this suggestion, we now start every section of the results with a sentence summarizing the most important findings.

2. *The results should present a sensitivity analysis of the parameters that significantly affect the model output, which aids in understanding which parameters are most critical to the model's outcomes.*

This analysis has been performed, and is described in section 2.5.1. The results are briefly presented in section 2.3.2 and the results shown in Fig. S5. We hope this is sufficient for the reader to visualize and understand the effect the different parameters have on the model outcomes.

[**answer**]

3. *If there are limitations to the results, such as the representativeness of the data or the applicable conditions of the model, they should be clearly stated in the Results section.*

The aim of the developed model for the presented manuscript was not to promote its application to a specific environment. Instead, we developed a model in line with the current knowledge of the SOC cycle similar to other recently developed models, with the aim of showing how equifinality can affect simulations of the turnover rate (through $\Delta^{14}C$) of SOC. To make this more clear to the reader, we added the following sentence to the first paragraph of the methods section: "*We note that the main aim of the developed model for the present manuscript was to show the effect of equifinality on model outcomes, and that the application of SOILcarb to other environments requires further testing.*".

**Discussion**

1. *The discussion should be expanded to address the generalizability of the model results, specifically whether the model is applicable to other soil types or environmental conditions, with further explanation provided in the discussion.*

As explained in the previous response, our aim was not to develop a SOC model that is widely applicable (although it has the potential to be, given that certain changes are made, for example, to apply it to agroecosystems). Therefore, a discussion on this is beyond the scope of our discussion.

To make the general application of our results more clear to the reader, the first sentences of section 4.2 ("Overparameterisation and equifinality in soil biogeochemical models") has been changed to "*Our result show that overparameterisation, which arises when a numerical model has too many parameters compared to the data available to constrain parameter values, has important consequences for the correct simulation of SOC dynamics. As many of the recently developed SOC model have a similar structure and use similar equations, it is likely that this is a general issue for such models, as has previously been shown for conventional turnover-based pool models (Braakhekke et al., 2013; Luo et al., 2016, 2017).*".

2. *The relative importance of different mechanisms at different soil depths can be further explored regarding its causes and potential influencing factors.*

It is not exactly clear which mechanisms at different soil depth could be more explored. In section 3.4 we discuss how different model aspects affect the $\delta^{13}C$ depth profile in the top and subsoil. We hope this provides sufficient information to the reader about how this novel aspect in the model (the simulation of $\delta^{13}C$ depth profiles) is affected by different model parameters.

3. *Although the article mentions the importance of accurately simulating the turnover time of SOC for predicting changes in the global carbon cycle, it can further expand the discussion on the specific significance of the research results in practical applications, such as the potential impacts and inspirations on soil management, climate change response strategies, and other aspects.*

To make the practical applications of the results more clear to the reader, we have added the following to section 4.3 ("Ways forward to identify and reduce equifinality in microbially-driven SOC models"): "*This is particularly important as these models are incorporated in Earth system models to make predictions of the response of the SOC stock to changes in the Earth's climate (e.g., Wieder et al., 2024), or to assess how changes in agricultural management practices can increase the amount of SOC to mitigate climate change and assign carbon credits (e.g., Mathers et al., 2023).*".

4. *Although the author mentioned that the model does not include the effects of temperature and soil moisture, it is suggested to further discuss the specific impacts of these limitations on the model's predictive ability.*

As noted above, the aim of the study was not to develop a model that can be used for predictions under changing environmental conditions, or can be readily applied to a wide range of environments. We do believe that this is possible, after certain modifications are made to the model structure (for example, the simulation of the soil N cycle and a coupling to a plant growth module). Therefore, it is beyond the scope of the discussion to provide information on this model limitation.

**Minor revisions**

1. *In Figure 3, the "calibrated for C" section is in italic format; please consistent formatting.*

Thanks for noticing, all similar labels have been changed to italic format.

2. *There are several sentences in the article that are rather cumbersome, such as the following ones:*

- *The model first calculates fluxes of 12C between pools and subsequently uses the ratio of 12C leaving every pool to the total amount of 12C of the respective pools to calculate how much 13C and 14C leave every pool, based on the respective 13C/12C and 14C/12C values of the pools. The model parameters are thus defined based on the 12C content of every pool.( Line 135 )*

Thanks for pointing this out. We rephrased this sentence and provided an equation to show what we mean.

- *The parameter sets to calculate the conditional and unconditional CDFs were obtained using the Matlab® version of SAFE toolbox (Pianosi et al., 2015), which was also used to post-process the results and calculate the sensitivity of the tested parameters using the Kolmogorov - Smirnov (KS) statistic. ( Line 282 )*

To make this sentence more comprehensible to the reader, we added the following to the first part of this paragraph: "[...] *This is a density-based global sensitivity analysis that quantifies the model sensitivity related to uncertainties of input parameters based on the cumulative distribution function (CDF) of the output distribution. This is done for the CDF when all parameters are varied (the unconditional CDF) and when one parameter is kept constant (the conditional CDF). The distance between both cumulative distributions is used to quantify the sensitivity of model to different parameters, and is calculated using the Kolmogorov-Smirnov (KS) statistic.*".

- *Similarly for the soil, after parameters are optimised using measurements of depth profiles of OC, δ13C and Δ14C of the POC and MAOC pools, simulated depth profiles of these fractions closely reproduce measurements. (Line 307).*

We rephrased this as follows: "*Similarly for the soil, after parameters are optimised using measurements of depth profiles of OC, $\delta^{13}$C and $\Delta^{14}$C of the POC and MAOC pools, the measurements of both pools are simulated very well by the model (Fig. 2).*".

---

## Author Comment (AC3)

Feedback from the reviewers is written in italic face, while our responses are written in normal font in green.

**Reviewer 3**

*The authors present a new mechanistic model (SOILcrab) to assess the potential to decrease parameter equifinality by including isotope data of tow soil fractions as constraints on parameter values during model optimization. They found that adding $\Delta^{14}C$ data as a calibration constraint, can correct simulation of the turnover rate of SOC and only substantially reducing equifinality for the parameter regulating desorption rate of OC from minerals. However, adding $\delta^{13}C$ data had little effect to improve simulations of the turnover rate of SOC or limit parameter equifinality. These findings are interesting and can improve the predictions of soil carbon dynamics under environmental change scenario.*

*Major concern:*

*This model was only applied in a deciduous forest site with different soil profile, it is unclear what's the performance of this model when it is applied to a large spatial scale.*

We thank the reviewer for taking the time to read our manuscript and provide feedback. As we responded to another reviewer (see above for reviewer 2), the aim of the model for the presented manuscript was not to develop a model that is readily applicable to a wide range of environments under different conditions, but rather to construct a model with a structure similar to other recently-developed microbially-driven SOC models to show how equifinality influences model results. Therefore, it is beyond the scope of the discussion section to address this issue, and leave this for future research as we apply the model (or an adapted version) to other environments.